Data-driven discovery of the spatial scales of habitat choice by elephants

Mashintonio Andrew F. 1 andrmash@scarletmail.rutgers.edu
Pimm Stuart L. 2
Harris Grant M. 3
van Aarde Rudi J. 4
Russell Gareth J. 1 5
1 Department of Biological Sciences, Rutgers University , Newark, NJ , USA
2 Nicholas School of Environmental Science, Duke University , Durham, NC , USA
3 United States Fish and Wildlife Service , Albuquerque, NM , USA
4 Conservation Ecology Research Unit, University of Pretoria , Pretoria , South Africa
5 Department of Biological Sciences, New Jersey Institute of Technology , Newark, NJ , USA
Perez-Acle Tomas
Electronic publication date: 2014 Aug 19
Publication date: 2014
Volume: 2
Electronic Location ID: e504
Received 2014 May 28; Accepted 2014 Jul 15
Copyright: © 2014 Mashintonio et al.
Copyright year: 2014
Copyright holder: Mashintonio et al.
License: This is an open access article distributed under the terms of the Creative Commons Attribution License, which permits unrestricted use, distribution, reproduction and adaptation in any medium and for any purpose provided that it is properly attributed. For attribution, the original author(s), title, publication source (PeerJ) and either DOI or URL of the article must be cited.
License URL: https://creativecommons.org/licenses/by/4.0/

Keywords: Etosha National Park, Loxodonta africana, Maputo Elephant Reserve, Resource selection function, Scale-dependent preference, Smoothing kernel

Funding: US Fish & Wildlife Services 98210-2-G365 98210-3-G651 98210-2-G300 Peace Parks Foundation PPF/P/24 The fieldwork was funded through grants from the US Fish and Wildlife Service (98210-2-G365, 98210-3-G651 & 98210-2-G300) and the Peace Parks Foundation (PPF/P/24) to RJ van Aarde. The funders had no role in study design, data collection and analysis, decision to publish, or preparation of the manuscript.

==============================
Setting conservation goals and management objectives relies on understanding animal habitat preferences. Models that predict preferences combine location data from tracked animals with environmental information, usually at a spatial resolution determined by the available data. This resolution may be biologically irrelevant for the species in question. Individuals likely integrate environmental characteristics over varying distances when evaluating their surroundings; we call this the scale of selection. Even a single characteristic might be viewed differently at different scales; for example, a preference for sheltering under trees does not necessarily imply a fondness for continuous forest. Multi-scale preference is likely to be particularly evident for animals that occupy coarsely heterogeneous landscapes like savannahs. We designed a method to identify scales at which species respond to resources and used these scales to build preference models. We represented different scales of selection by locally averaging, or smoothing, the environmental data using kernels of increasing radii. First, we examined each environmental variable separately across a spectrum of selection scales and found peaks of fit. These ‘candidate’ scales then determined the environmental data layers entering a multivariable conditional logistic model. We used model selection via AIC to determine the important predictors out of this set. We demonstrate this method using savannah elephants (Loxodonta africana) inhabiting two parks in southern Africa. The multi-scale models were more parsimonious than models using environmental data at only the source resolution. Maps describing habitat preferences also improved when multiple scales were included, as elephants were more often in places predicted to have high neighborhood quality. We conclude that elephants select habitat based on environmental qualities at multiple scales. For them, and likely many other species, biologists should include multiple scales in models of habitat selection. Species environmental preferences and their geospatial projections will be more accurately represented, improving management decisions and conservation planning.

Introduction

Successful species conservation and management requires understanding the resources needed for their reproduction and survival (see Roever, van Aarde & Legget, 2012; Roever et al., 2013; Roever, van Aarde & Chase, 2013). Because some resources are difficult to identify directly, habitat preferences can serve as proxies (Young, Ferreira & van Aarde, 2009). They, in turn, are revealed by the locations and movements of individuals within their landscape (Manly et al., 2002; Aarts et al., 2008; McLoughlin et al., 2010; Fisher, Anholt & Volpe, 2011; Roever et al., 2013). Models of habitat preference usually incorporate raster-based information, such as vegetation maps, at a spatial resolution determined by the data source (e.g., satellite imagery). This practice assumes that animals judge habitats at the same level of detail, or scale. However, organisms may respond to more fine-grained variation, or coarser, aggregated qualities, depending on the spatial context or their perceptual ability (Holling, 1992; Lima & Zollner, 1996; Nams, 2005; de Knegt et al., 2010; Marshal et al., 2011). In fact, the resolution of the data may be biologically irrelevant for the species in question, which can limit model inference and produce potentially misleading results (Levin, 1992; Boyce, 2006; Mayor et al., 2009; Wheatley & Johnson, 2009; de Knegt et al., 2011).

A priori, biologists rarely know the spatial scale at which species select resources. Further, there is evidence that for some organisms, a single “characteristic” scale (Holland, Bert & Fahrig, 2004; de Knegt et al., 2010) may inadequately characterize an environmental response (Mayor et al., 2009; Wheatley & Johnson, 2009; Fisher, Anholt & Volpe, 2011; Shrader et al., 2012). Here, we demonstrate how to identify the most important scale(s) of habitat selection by examining relationships between species movements and environmental attributes over a continuum of scales. We show that this data-driven approach changes the predictions of the amount and distribution of suitable areas across the landscape.

For a human example of multiple preference scales, imagine a suburban family that enjoys shopping. In the suburbs, stores are aggregated in a characteristic way, with high local concentrations (plazas, malls, etc.) separated by areas with few or no stores. Most of the area in Fig. 1A, in which dark grid squares represent high store density, has the suburban pattern. The path of the family’s travels–the black line–clearly shows that shopping areas are frequent targets. An analysis focusing only on the suburbs would reveal a preference for high store density.

Figure 1 Example of how spatial scale can affect preference.

(A) Hypothetical store density map of a city and surrounding suburbs, where dark grid squares represent high store density, overlaid with the movement path of a suburban family. Local movements indicate a preference for stores, but this preference does not extend to the city, where shopping opportunities are abundant. (B) Store density map after smoothing with a 21-pixel Gaussian filter. Now, it is apparent that the family selects against store density at a larger scale, even though it selects for store density at a smaller scale.

Next, consider the area in the lower right corner. Knowing only store data, we would rightly guess that this is a city. A naïve extrapolation of the family’s suburban movements would predict frequent visits to this city, where stores are abundant. However, we would be completely wrong; our hypothetical family avoids cities. They do so because despite attractive qualities, such as high store density, cities have perceived disadvantages: crowds, lack of parking, and so on. These attributes only become important when store density is assessed at a larger scale than that of a suburban mall. The key point is that the data describing store density serves as a proxy for different qualities at different scales. To unravel this, we can locally average, or smooth, these data with an increasingly larger radius. Figure 1B shows the store data smoothed using a 21-pixel Gaussian filter. This converts the landscape to a map of large-scale urban density, and we can interpret the family’s travels as “avoiding the city”. By including both the original density map and the smoothed version in a model, we simultaneously discover the preference for isolated stores and the avoidance of large aggregations of stores. Even if we did not know exactly why this family avoids high store densities, our predictions of their future travels will be more accurate.

A previous study on savannah elephants from Maputo Elephant Reserve, Mozambique and Etosha National Park, Namibia, incorporated travel costs with other habitat variables to generate landscape-wide quality maps (Harris et al., 2008). They determined habitat preferences using variables at a 500 m resolution, which is very detailed given that elephants can move across thousands of square kilometers within a year. These models were able to accurately predict local movement choices, that is, the places that elephants chose over their immediate neighbors in the areas where they had been observed. However, their ability to provide regional predictions might break down when extrapolated over a broader landscape, such as the entirety of a reserve (see also Roever, van Aarde & Legget, 2013). By analogy to the store example, elephants in more open savannah might tend to stay near trees while avoiding large forests.

Using the same dataset as the earlier study, we tease apart these scale-dependent preferences by smoothing each of the original environmental variables at different radii and assessing how well each explains animal movements. (Operationally, we define “scale” as the width of the radius used to smooth the original environmental data, so scale 0 refers to the original data.) All of the variables, each at one or more identified optimal scales, are then used in a model selection process to generate a final landscape preference model. While our multi-scale models agree with the previous findings that elephants prefer to occupy areas that are near water, have high vegetation cover, and are far from human settlement, they predict local movements much better than models that use only a single scale. The multi-scale models can also produce very different predictions of landscape-wide habitat quality, potentially improving conservation directives that aim to protect essential habitats.

Materials and Methods

Study sites

Maputo Elephant Reserve and the Futi River corridor, which extends south of Maputo and is also included in the analysis, are located in the subtropical savannahs of southern Mozambique. At the time of the study, the reserve (c. 800 km2) was unfenced except for a 30 km stretch in the northwest (Harris et al., 2008). At least 311 elephants lived in the reserve and the corridor when these data were collected (Olivier, Ferreira & van Aarde, 2009). Etosha National Park (c 23,000 km2) is located in the arid north-central part of Namibia. This park was fenced and held approximately 2,000 elephants at the time of the study (de Beer et al., 2006).

Location data

GPS collars provided elephant location data (held by CERU, www.ceru.up.ac.za/). In Maputo they provided fixes every two to five hours, with collars active for 24 h and off for 48 h (Harris et al., 2008). Three males and two females wore collars covering the wet seasons (November–March) of 2000 and 2001 and the dry seasons (April–October) of 2001 and 2002. In Etosha, location data from six females were taken every eight hours and encompassed the wet seasons of 2002 and 2003 and the dry season of 2003 (Harris et al., 2008). Each female that was collared represented the movements of an entire herd. The data collection was facilitated through permission from the Namibian Ministry of the Environment (Research Permit number 580).

Individual movement patterns and habitat selection vary with sex (Stokke & du Toit, 2000; Woolley et al., 2009) and season (Wittemyer et al., 2007; Young, Ferreira & van Aarde, 2009; Young & van Aarde, 2010). Therefore, we combined location data in each reserve separately for males and females during the wet and dry seasons. Analyses were seasonal in resolution, so we did not partition movements by time of day. This yielded four data sets for Maputo and two for Etosha (for which only females carried collars). A pair of x, y coordinates represented the starting and ending location of each movement. We considered only those movements within a choice radius of <5 km, because fast, long-distance movements may carry a different signal of habitat selection than slower, shorter movements (Morales et al., 2004; Roever et al., 2013). This process retained >80% of the movements in each dataset.

Landscape data

The landscape variables consisted of vegetation, distance to water, and distance to human settlement. In Maputo, vegetation data included the proportion of reeds and tree cover. In Etosha, the vegetation variables included the proportion of mopane, Acacia nebrownii, and Acacia-dominated savannah (henceforth, Acacia). All raster-based variables had a cell, or pixel size, of 500 m by 500 m. We standardized each variable to have zero mean and unit standard deviation across the entire landscape.

We created squared versions of each variable and included these in the smoothing and model selection process where appropriate (see below). This allows for a variety of non-linear preference functions, including those in which an intermediate level of a variable is preferred (Johnson, Seip & Boyce, 2004; Johnson & Gillingham, 2005). This possibility is likely to be important if animals integrate their surroundings. For example, imagine an animal that likes open spaces in a savannah habitat, a mosaic of trees and grassland. At the fine scale, it might show a monotonic preference for open space, but at a larger scale, it would prefer the intermediate level of tree cover that characterizes a savannah. Expanding on this example, Fig. 2 describes possible interpretations of different combinations of preference function shapes at different scales.

Figure 2 Possible interpretations of certain combinations of selection functions at small, medium, and large scales.

Small-scale refers to smoothing <5 pixels, medium-scale refers to smoothing up to the choice radius (10 pixels), and large-scale refers to smoothing greater than the choice radius. Point and linear features are coded as ‘distance to …’ arrays, which are intrinsically smooth at scales up to the typical distance between features, and so are analysed without further smoothing.

Smoothed landscape data

We created smoothed vegetation variables by averaging each pixel with its neighbors within increasing radii up to 20 pixels, i.e., 10 km (Fig. 3). The functional form of the smoothing kernel should approximate the way an individual integrates its surroundings. For example, a flat smoothing kernel, where all pixels are weighted equally within a given radius, would be appropriate if an individual’s ability to assess information remained constant within that radius, and then dropped off dramatically (e.g., Holland, Bert & Fahrig, 2004; de Knegt et al., 2011). Equally plausibly, an individual’s perceptual abilities might decline gradually with distance. This could be caused by any number of mechanisms, such as diffusion and decay of chemical signals or progressive visual obstruction by vegetation. To account for this range of possibilities we included a linear decay parameter d, which affects the weight given to each pixel in the average depending on its distance from the central pixel. For the flat kernel, d = 0 (Fig. 4A). We allowed d to increase in steps of 0.1 (Fig. 4B) up to a maximum of d = 1, where the smoothing kernel declines linearly to 0 at the edge of the radius (Fig. 4C). This parameter was optimized along with the radius (see below).

Figure 3 Proportion of reeds in Maputo smoothed at increasing scales.

The original (base-scale) landscape is on the left, followed by landscapes smoothed at 1, 5, 10, and 20 pixels.

Figure 4 Smoothing kernels with varying decay.

(A) A flat smoothing kernel, where all pixels within the radius are averaged equally. (B–C) Decaying smoothing kernels, where pixels closest to the central pixel are weighted more heavily in the average than pixels that are farther away.

We did not smooth ‘distance to’ variables because they are intrinsically smooth. We did create a squared version of ‘distance to water’ along with a squared version for each vegetation variable at each scale.

When smoothing, we can treat landscape values outside the spatial extent of the available data in one of two ways: either as true zeroes that represent habitat unsuitable for the organism (e.g., an ocean for a terrestrial mammal), or as unknown values. There may also be physical boundaries, such as fences, within the areas under consideration. In the case of a fence, while an animal may be unable to visit a location outside the fence, it is unclear whether it will take into account the bordering habitat when choosing a location inside the fence. In our study, fences coincided with the border of our landscape data in parts of Maputo and all of Etosha. While we may not know the habitat immediately bordering the reserve, there is no reason to think it is radically different from what is inside. We therefore smoothed using the average of only those pixels within the smoothing radius that are also within each reserve, generating a ‘neutral’ boundary.

Habitat selection model

Resource selection functions (RSFs) specify the probability that a particular resource (or in this case, habitat) is chosen by an animal (Manly et al., 2002). These functions have been increasingly used to assess habitat selection based on movement data (i.e., McLoughlin et al., 2010), especially for elephants (Roever, van Aarde & Legget, 2012; Roever et al., 2013; Roever, van Aarde & Chase, 2013; Roever, van Aarde & Legget, 2013). For input they require a set of movement data and a set of landscape variables (or ‘layers’) describing the environment in which the organism(s) are moving.

Except for the most mobile animals, not all habitats can be reached in a given time interval, such as the 8 h GPS fix interval. Therefore, only a subset of habitats within a certain radius of an individual are even candidates for being ‘chosen’ (Arthur et al., 1996). For any given movement i, each potential destination pixel j has a vector of k potential predictor values xj, derived from the landscape raster layers. Included in these values is a distance term between the current location and each potential destination pixel, which represents the cost of movement (Hjermann, 2000; Harris et al., 2008). The actual choice yj is represented as a binary response, where the chosen location is given a value of 1 and all other locations, or a random subset of them if there are too many, are given a value of 0. Thus, the complete dataset for a single movement consists of a matrix Xi and a binary column vector yi, in our case covering the chosen destination pixel and 29 random alternative destinations in a 10-pixel circular neighborhood of the starting pixel.

Under the conditional logistic model, the probability pj that an animal will choose a pixel j as its next location is pj=exjβ∑jexjβ

where β is a k by 1 vector of parameters to be estimated. Li, the log-likelihood of a particular movement, is simply the logarithm of the value of pj for the chosen destination pixel (the case where yj = 1). The log-likelihood of the entire data set is the sum of the Li over all movements i. We fit the model using non-linear maximization of log-likelihood using a quasi-Newton method implemented in the Mathematica software package (Wolfram Research, 2012), and compared the models’ performance using Akaike’s Information Criterion (AIC).

Harris et al. (2008) found that the estimated model parameter applying to the ‘travel distance’ variable was extremely consistent across elephants and varied very little with the inclusion of other landscape variables. We followed their suggestion by first fitting that term separately and then using the fitted value as a fixed term when optimizing the other parameters.

Choosing optimal smoothing scales

There are two steps in the model discovery process. The first is the identification of candidate smoothing scales for each variable. The second is the inclusion of all the candidate variables (original or smoothed) in a model selection process.

To identify candidate scales, we smoothed each variable independently within increasingly larger radii. The radius extended from 0 pixels (the original data) to a maximum of 20 neighboring pixels (10 km), which is twice the radius of the local neighborhood of movement choices. (In initial runs we encountered an issue with over-smoothing if the radius was too large, which produced unusual results; see Discussion.) At each radius we fit three models: one that included only the distance from current location, one with distance and the smoothed variable, and one that also included the smoothed variable squared. In each case we recorded the AIC score, generating three lines of AIC values (one of which, for distance only, is a constant). Candidate scales were chosen by looking for minima in the AIC lines. (In the figures, we invert the AIC axis so that the best models are peaks.) After identifying the candidate scales, we then optimized the decay parameter for each scale.

An alternate approach would be to smooth all landscape variables together at the same scales, instead of independently (Fisher, Anholt & Volpe, 2011). This would be appropriate if, for example, a single constraint determined the manner in which organisms integrate their environment, such as their perceptual ability or physiology, which applied equally to all variables (Lima & Zollner, 1996). We tested this but found that after the final model selection process (see below), the ‘separate scales’ model always equaled or outperformed the ‘same scales’ model, so we did not continue this analysis. The reason seems to be that when we smooth all variables together, the smoothing profile will typically be dominated by the variable that has most impact on the likelihood values. The best scales for the other variables remain hidden.

Choosing the final model

After identifying the candidate smoothing scales for each variable, we entered the linear and/or quadratic versions of these variables, distance from current location, distance from water, distance from human settlement (if applicable), and any original variables that had optimal AIC scores into a model selection process. We used AIC to determine the most parsimonious model, and to create a parameter-averaged model based on AIC weights (Burnham & Anderson, 2002). For comparison (see below), we repeated the model selection process, only allowing ‘base’ (original, non-smoothed) variables (as per Harris et al., 2008).

Two kinds of predictions

We created two types of habitat quality maps using the parameter-averaged model. The first type of map has quality values given by pj, which measure relative local quality. Since this is the basis for the conditional logistic model, these maps illustrate model fit. We only show the predictions for the neighborhoods surrounding the start of each observed movement. Each map was then overlaid with movement end-points to show how well these coincide with predictions of locally optimal locations.

We also calculated the mean deviation of the probability value of each chosen pixel from all other pixels in its local 10-pixel neighborhood. Large positive values indicate that a high-quality pixel (preferred habitat based on model prediction) was chosen out of a variety of options, large negative values indicate that a low-quality pixel was chosen, and intermediate values mean either that a medium-quality pixel was chosen, or that there was very little variation (the landscape was relatively uniform). To assess the impact of including multiple scales of preference, we created local prediction maps and calculated mean deviations for the base-scale models and compared them to the multi-scale models using histograms.

The second type of map has quality values given by exjβ, a measure of relative landscape-wide quality. These values were calculated for the entire landscape (an extrapolation). As above, each map was overlaid with movement end-points, this time to assess how well they predict the landscape-wide distribution of the elephants. Equivalent maps that do not allow smoothed variables were also created for comparison. Figure 5 shows the complete process for one dataset: male elephants in Maputo during the dry season.

Figure 5 Process of model selection with multiple scales.

(A) Identify smoothing radii. The smoothing radii for the variables reeds and tree cover were optimized separately for three models: distance from current location only (flat solid line), distance and habitat variable (jagged solid line), and distance with both the linear and quadratic habitat variables (dashed line). Each of the peaks of model fit at the various radii is indicated. (B) Identify smoothing decay. The decay was optimized for each of the optimal radii in (A). The optimal decay for each radius is indicated. (C) Create landscape variables. Maps were created for each of the variables at the optimal radius and decay. Each map is a composite of the linear and quadratic values. (D) Find best model. The overall model choice uses the distance from current location, each of the selected variables at the optimal radius and decay, distance to water, and distance to human settlement (if applicable) as input parameters. The best model was chosen as the combination of model parameters with the lowest AIC score; in this case, the score was 3,472.33. The importance of the parameter is measured from the weights of the models in which it appears, and the parameter-averaged value is the value of the parameter averaged across all models. (E) Final predictions. The local relative quality map was created using the parameter-averaged values for all of the model parameters and applied to the 10-pixel radius of local movement choices for each start point (top). The map was overlaid with the endpoint of each movement to assess elephant choice. The landscape-wide relative quality map was created using the parameter-averaged values for all of the model parameters and applied to the entire landscape (bottom). The map was overlaid with the endpoint of each movement to assess elephant choice.

Results

Multiple scales and model performance

For all elephant groups at both reserves, model fit peaked at distinct smoothing radii (scales) for different habitat variables when considered individually. In many cases, a variable showed multiple peaks at different scales (Fig. 6; Table 1). After multivariate model selection using these scales, in each of our six datasets, the best multi-scale model was more parsimonious than the corresponding best base-scale model, according to AIC (Table 2). The difference in AIC score for Maputo datasets ranged from 31.0 to 48.5. In Etosha, the dry season models differed by 15.6, the smallest difference between any two models; this is evident in the similarities of the landscape-wide preference maps (see Fig. 8). The difference between the wet season models is 68.9; this also corresponds to the most striking difference between the preference maps (see Fig. 8). Overall, the results indicate that these elephants are using aggregated habitat attributes when deciding where to move, but in a different way depending on the season, the reserve, and the sex of the elephants.

Figure 6 Habitat selection by female elephants in Etosha for the variables mopane, A. nebrownii, and Acacia in both the wet and dry seasons.

(A–C) During the wet season, when water is not limiting, individuals utilize more of the landscape and select habitat variables at larger scales more strongly than in the dry season. (D–F) During the dry season, when individuals are more restricted in their movements to areas near water, habitat variables are typically selected more strongly at a smaller scale than at a larger scale. Each variable is fit to three models: distance from current location only (straight solid line), distance and the linear habitat variable (jagged solid line), and distance with both the linear and quadratic values (dashed line). Each of the peaks of model fit is indicated. A single asterisk indicates that only the linear value was chosen, and a double asterisk indicates that both the linear and quadratic values were chosen. An asterisk in parentheses indicates that the variable was not included in the best overall model.

Table 1 Candidate scales and decay of each variable chosen for the model selection process for each dataset and their estimated parameter values after model fitting.

Variable	Decay	Best model
parameter
value	Importance	Parameter-
averaged
value	Variable	Decay	Best model
parameter
value	Importance	Parameter-
averaged
value	
Maputo females, dry season	Maputo females, wet season	
Distance	–	−1.10	1.00	−1.10	Distance	–	−0.97	1.00	−0.97	
Reeds 4	0.0	0.63	0.97	0.63	Reeds 2	0.0	0.14	0.81	0.32	
Reeds 42	0.0	−0.15	0.94	−0.15	Reeds 22	0.0	–	0.58	−0.06	
Reeds 20	0.0	2.02	0.80	1.87	Reeds 18	0.0	–	0.39	0.59	
Reeds 202	0.0	−2.57	0.99	−2.12	Reeds 182	0.0	–	0.32	−0.24	
Trees 1	0.0	0.34	0.64	0.30	Trees 0	–	0.36	1.00	0.34	
Trees 12	0.0	−1.00	1.00	−1.08	Trees 02	–	−0.25	1.00	−0.24	
Trees 8	0.7	−1.80	0.53	−1.11	Trees 14	0.7	−0.72	0.52	−0.65	
Trees 82	0.7	–	0.31	−0.30	Trees 142	0.7	−3.20	1.00	−2.94	
Trees 14	1.0	1.97	0.43	1.13	Water distance	–	−0.35	0.79	−0.33	
Trees 142	1.0	–	0.43	−1.05	Water distance2	–	–	0.31	−0.09	
Water distance	–	–	0.28	−0.03	Settlement distance	–	–	0.29	0.04	
Water distance2	–	–	0.28	−0.11						
Settlement distance	–	–	0.37	0.09						
Maputo males, dry season	Maputo males, wet season	
Distance	–	−1.07	1.00	−1.07	Distance	–	−1.04	1.00	−1.04	
Reeds 4	0.0	0.81	0.98	0.74	Reeds 0	–	–	0.33	−0.00	
Reeds 42	0.0	−0.34	1.00	−0.33	Reeds 02	–	–	0.48	−0.48	
Trees 1	0.0	0.52	1.00	0.47	Reeds 3	1.0	–	0.35	0.10	
Trees 12	0.0	−0.12	0.84	−0.11	Reeds 32	1.0	−0.06	0.46	−0.06	
Trees 10	0.9	−1.41	0.62	−1.25	Trees 2	0.0	0.44	1.00	0.43	
Trees 13	0.8	2.35	0.79	1.87	Trees 9	1.0	–	0.43	0.45	
Water distance	–	0.32	0.91	0.33	Trees 92	1.0	−0.81	0.99	−0.81	
Water distance2	–	−0.18	0.99	−0.18	Trees 15	0.7	–	0.32	−0.51	
Settlement distance	–	–	0.51	0.11	Trees 20	0.6	3.21	0.99	3.13	
					Trees 202	0.6	1.06	0.53	1.05	
					Water distance	–	−0.13	0.66	−0.12	
					Water distance2	–	–	0.32	−0.00	
					Settlement distance	–	–	0.28	−0.01	
Etosha females, dry season	Etosha females, wet season	
Distance	–	−1.13	1.00	−1.13	Distance	–	−1.39	1.00	−1.39	
Mopane 1	0.0	0.50	1.00	0.50	Mopane 1	0.8	0.16	0.85	0.14	
Mopane 7	0.0	0.23	0.91	0.24	Mopane 12	0.8	−0.17	0.97	−0.18	
A. nebrownii 1	0.0	0.21	1.00	0.21	Mopane 15	0.0	–	0.35	0.20	
A. nebrownii 18	0.0	–	0.30	0.09	A. nebrownii 0	–	0.35	0.98	0.30	
A. nebrownii 182	0.0	0.41	1.00	0.40	A. nebrownii 02	–	−0.07	0.87	−0.06	
Acacia 2	1.0	0.18	1.00	0.18	A. nebrownii 7	0.0	−0.60	0.85	−0.64	
Water distance	–	−0.21	0.57	−0.24	A. nebrownii 72	0.0	0.23	0.84	0.21	
Water distance2	–	0.18	0.90	0.21	A. nebrownii 14	0.0	−0.71	0.81	−0.72	
					Acacia 2	0.0	0.57	0.99	0.59	
					Acacia 10	0.0	1.46	0.52	1.31	
					Acacia 102	0.0	3.11	0.84	2.72	
					Acacia 20	0.0	−9.20	1.00	−8.02	
					Acacia 202	0.0	−11.59	0.99	−9.83	
					Water distance	–	−0.17	0.48	−0.19	
					Water distance2	–	–	0.30	−0.01	

Table 2 AIC scores for the best model in each dataset for both the multi-scale and the base-scale versions.

Dataset	Multi-scale	Base-scale	Difference	
Maputo females, dry season	2901.03	2944.45	−43.42	
Maputo females, wet season	2748.15	2779.19	−31.04	
Maputo males, dry season	3472.33	3520.81	−48.48	
Maputo males, wet season	4112.84	4155.37	−42.53	
Etosha females, dry season	11545.80	11561.40	−15.60	
Etosha females, wet season	9439.26	9508.14	−68.88	

The shapes of the relationships

Each elephant group displayed a quadratic preference relationship with at least one habitat attribute at the candidate scales, e.g., Acacia at 20 pixels (Fig. 6C). In total, 19 quadratic variables were chosen in the second, model selection step for the multi-scale models, and 8 were chosen for the base-scale models (Table 1; see Table S2 for base-scale model fits). By examining the shapes of these quadratic functions over the range of habitat occupied by individuals (icons on Fig. 7), we see that in 10 cases the squared term specifies a curvilinear but still essentially monotonic relationship. In 13 cases, a unimodal ‘hump’ is observed (e.g., Maputo females during the dry season for tree density at 1 pixel, or 0.5 km). In 4 cases, we observed a function with an intermediate minimum (e.g., Etosha females during the wet season for A. nebrownii at 7 pixels, or 3.5 km). These fitted functional forms guide our interpretation of the behavioral ecology of the elephants (Fig. 2).

Figure 7 Local relative quality maps.

The maps were overlaid with the endpoint of each movement for all elephant data sets, representing the composite of all variables present in the best multi-scale model and the original, base-scale variables only. The histograms show the comparison between the mean deviation of the probability values of the multi-scale maps (white) and base-scale maps (black). The variables at their optimal scale(s) are shown with the shape of the elephant’s relationship to each variable.

Figure 8 Landscape-wide relative quality maps.

The maps were overlaid with the endpoint of each movement for all elephant data sets, representing the composite of all variables present in the best multi-scale model and the original, base-scale variables only.

Variables and scales describing elephant preferences

• Distance. The distance parameter was always negative and varied from −0.97 to −1.39. Taking into account the different numbers of available pixels in different distance bands, this corresponds to probabilities of moving 0, 0.5, 1, or >1 km from the original location of 0.15, 0.37, 0.22, and 0.27 (for parameter −0.97) and 0.28, 0.44, 0.17 and 0.11 (for parameter −1.39). When elephants move locally they prefer to move as little as possible in these reserves.

• Reeds, Maputo. When smoothing the reed data, for each elephant group except males in the wet season, a flat kernel (d = 0) was optimal for reeds regardless of scale, indicating that all pixels contributed equally. Males in the wet season had a decaying kernel (d = 1.0) for reeds at a scale of 3 pixels, indicating pixel influence decreased with distance from the central pixel. The model selection results reveal that at small scales, males avoided reeds in the wet season (3 pixels, or 1.5 km) but had an intermediate preference (maximum at 38% cover) in the dry season (4 pixels, or 2 km). Females had a small-scale preference for reeds in both seasons. However, only females during the dry season had a preference for reeds at the large scale (20 pixels, or 10 km), which was highest at 20% cover. Reeds occur in large stretches that appear homogenous from a distance (i.e., larger scale), but there are small openings within the beds (i.e., smaller scale) where elephants bathe and drink. This may explain why most elephant groups only have a relationship with reeds at a small scale. Reeds indicate the presence of consistently wet areas, so it is not surprising that reedy areas are more attractive for all elephants in the dry season, when water is scarcer elsewhere.

• Trees, Maputo. For all elephants in Maputo, a flat kernel provided the best fit for smoothing of tree cover at small scales (up to 2 pixels, or 1 km). At larger scales (>8 pixels, or 4 km), decaying kernels with d from 0.6 to 1.0 were best (Table 1). This pattern suggests a relatively short perceptual range for tree cover. Following model selection, at small scales males in both seasons showed a positive preference for trees, but for females this was a convex quadratic function with an intermediate optimal tree density of 19% in the dry season and 28% in the wet season. This suggests the base resolution of environmental data used in this study (500 m) might be too large and is already blurring the savannah ‘mosaic’ in Maputo as perceived by females (see Fig. 2, pattern I). Males in both seasons also have a negative relationship with tree density at medium scales (9–10 pixels, or 4.5–5 km), although for the wet season there is a slight intermediate peak. This becomes a positive preference at even larger scales (>12 pixels, or 6 km). This combination suggests a preference for medium-sized clearings within continuous forest (Fig. 2, pattern C). Females in the dry season show this same alternating preference for tree density at medium and large scales. However, females in the wet season demonstrate a negative relationship for tree density with a slight intermediate peak at a large scale (14 pixels, or 7 km). Overall, these results suggest that elephants like to be near small clumps of trees within relatively open areas, with all groups but females in the wet season willing to venture into more continuous forest.

• Vegetation, Etosha. In both seasons, a flat smoothing kernel was optimal for each of the vegetation variables at all scales except in two cases: Acacia at a scale of 2 pixels in the dry season and mopane at a scale of 1 pixel in the wet season had decaying kernels (d = 1.0 and 0.8, respectively). Because these habitats are already at a small scale, this indicates that elephants only consider these vegetation types in their immediate vicinity when making habitat choices. Following model fitting, we find that female Etosha elephants preferred higher densities of mopane and Acacia at a small scale in both seasons. They also showed a preference for higher local densities of A. nebrownii in the dry season, but intermediate levels in the wet season (43% cover). At larger scales, in the dry season, the elephants favoured mopane at 7 pixels (3.5 km) and A. nebrownii at 18 pixels (9 km). In the wet season, they had a u-shaped relationship to A. nebrownii at 7 pixels (3.5 km) and a negative one at 14 pixels (7 km), indicating an avoidance of the edges of large patches of A. nebrownii. They also had a slight u-shaped relationship with Acacia at 10 pixels (5 km) but showed a preference for an intermediate amount of Acacia at 20 pixels (10 km). This suggests an avoidance of Acacia mosaics but a preference for being near the edge of them. (Note the full-landscape multi-scale map for the Etosha wet season in Fig. 8 shows a cluster of observations on the edge of the unfavorable blue area, which is an Acacia-dominated mosaic.) Seasonal differences in selection scale are also evident for these elephants: mopane and Acacia vegetation contributed much more strongly at a large scale (>15 pixels) in the wet season than in the dry season (compare Figs. 6A and 6C with 6D and 6F).

• Water. As a ‘distance to’ variable, the water layer is intrinsically smooth, so there is only one scale. In all Maputo datasets except one, elephants dislike being far from water. Water does not appear at all, however, in the best model for female elephants in the dry season, the dataset in which one might expect it to be the most important. However, Fig. 7 shows that all of our data for this period are from the north region of the reserve, which is the region with reeds, a surrogate for water (Harris et al., 2008). In the best model, reeds are positively preferred at a small scale (4 pixels, or 2 km) and intermediately preferred at a large scale (20 pixels, or 10 km), indicating females like to be on the edge of the reedy areas (Fig. 2, pattern E). Given that, we propose that water does not add much explanatory power compared to when elephants are further south, near the Futi corridor. We note that in the respective base-scale model, which does not allow a large-scale relationship with reeds, water is important as expected. In Etosha, elephants stay close to water during the wet season but have a u-shaped relationship during the dry season. For a point resource like a water hole, this suggests individuals are alternately moving to and away from it (Roever, van Aarde & Legget, 2013; Fig. 2, pattern K). This is because in the dry season, vegetation near water holes is rapidly denuded, forcing elephants to travel farther away from water to forage (de Beer et al., 2006). The respective base-scale model demonstrates the same relationship to water.

• Human settlement, Maputo. Elephants appear to pay little attention to human settlements in any dataset when fitting multi-scale models, but are important in the base-scale model for females in Maputo during the dry season. In all models in which ‘distance to human settlement’ appears, even models of low rank, its parameter is positive, indicating that, however mildly, these elephants avoided human settlements.

Comparison with base-scale models

For the most part, the vegetation variables that appeared in the best base-scale model for a particular dataset were the same as those that were in the best multi-scale model at a small scale. For example, for males in Maputo during the wet season, the best multi-scale model included reeds at 3 pixels and trees at 2 pixels; both reeds and trees were included in the best base-scale model. In one case, however, the shape of the relationship changed: females in Maputo during the dry season showed a positive preference for reeds at 4 pixels in the multi-scale model, but had a negative preference for reeds in the base-scale model. Additionally, there were two cases where the two types of models had different variables: reeds in Maputo for females during the wet season and males during the dry season were included in the multi-scale models (at 2 and 4 pixels, respectively) but not in the base-scale models.

Local relative quality predictions

These predictions form the basis of the model fit. Figure 7 demonstrates that the main difference between the multi-scale and base-scale models is that low-quality pixel errors—the left tail of each histogram—are much reduced when smoothed variables are incorporated (white bars) compared to when they are not (black bars). Without smoothing, many parts of these landscapes are locally heterogeneous, with high-quality and low-quality locations closely adjacent. Even when elephants cluster in areas with many high-quality pixels, they are inevitably sometimes found in the interspersed low-quality pixels, perhaps because they are moving between high-quality areas. When smoothing is allowed, these pixels increase in probability of occupation by virtue of their high-quality surroundings.

Landscape-wide relative quality predictions

The landscape-wide relative quality maps generated from the best multi-scale models are, in some cases, strikingly different from those based only on base-scale models (Fig. 8). As well as reducing low-quality pixel errors (see above), these maps sometimes reduce the total area predicted to be high quality that does not contain elephants. Whether elephants are intentionally avoiding these areas, or are present but not being observed, will require further data to test.

For Maputo, the landscape-wide maps that include smoothed variables show the regional distribution of elephants better than maps with only base-scale variables, as indicated by the areas of red pixels. Interestingly, the maps for males show their attraction to the north–south ‘tree corridor’ along the western side of the reserve, whereas the best models for females, when extrapolated into southern regions, suggest they would stay to the east. This difference arises mainly from their different relationships to tree density at various scales.

One exception to the pattern of multi-scale models making better predictions is for Maputo males in the wet season, when individuals sometimes venture into the northeast region of the reserve. This general area is assessed as relatively poor by the multi-scale model, whereas the base-scale model shows it as of medium quality with pockets of high quality. The multi-scale model does well elsewhere, where most observations occur, so presumably fit in those regions was favored.

For Etosha, the multi-scale and base-scale maps are similar in the dry season, and both classify regions where female elephants occur as high quality. The multi-scale map reduces the more minor of the two errors: there are fewer high-quality regions without elephants. For the wet season, the maps are very different. The base-scale map has mixed performance, with individuals found in regions classified from low quality (north-central and southern tip) through high quality. There are also extensive ‘high quality’ regions with no records of elephant presence. Overall, the base-scale map does not reflect the observed presence of elephants well. But once we allow smoothing the situation changes, and the multi-scale model indicates that (a) elephants seem to avoid the two Acacia-dominated regions (even though they like Acacia on a local basis), and (b) within the rest of the reserve, many large areas are suitable, allowing females to roam widely in a way that isn’t observed during the dry season.

Discussion

We use the data-driven smoothing approach presented here to identify the spatial scales at which an organism selects habitats. By smoothing each variable independently, we were able to identify its optimal scale(s) and improve model fit (Fisher, Anholt & Volpe, 2011). We add support to the proposition that organisms select habitat variables within the landscape at varying scales (Bowyer & Kie, 2006; Mayor et al., 2009). Selection depends on the spatial context of the variables (Duchesne, Fortin & Courbin, 2010).

Including squared values for variables in the model selection process allows for situations where an organism prefers an intermediate value of a particular variable (such as tree cover), or where it is moving towards and away from point features and/or avoiding edges of vegetation. These relationships occurred in our models, emphasizing the importance of using flexible resource selection functions in habitat analyses.

The predictions made by our base-scale models are in general agreement with those of Harris et al. (2008): elephants prefer to be close to water, within forage, and away from people. In the previous paper, the distance variables were the only smooth variables, possibly giving them greater predictive power compared to the other base-scale variables. In our multi-scale models, the signal of water preference is lost in some datasets, likely due to the inclusion of other, smoothed variables (like proportion of reeds) that signify water availability and possibly other landscape qualities. In two instances, distance to water is added to the best multi-scale model: males in Maputo and females in Etosha during the wet season. This is not evident in the landscape-wide quality maps, but there is still local variation in the quality of the habitat that is partly dependent on the distance to water.

Some habitat variables operate on multiple scales in opposing fashion, such as Acacia for females in Etosha during the wet season being selected positively at a small scale (2 pixels, or 1 km) but preferred in intermediate amounts at larger scales (20 pixels, or 10 km; Fig. 7). This indicates that individuals avoid regions dominated by Acacia (only selecting the edges) but favor isolated Acacia patches within other regions (Fig. 2, pattern E). But even at the smallest scale (for our study, a 500 m × 500 m area), the selection of a variable may be due to its association with a resource preferred at finer scales not captured by our data (e.g., presence of water). Additionally, in Maputo, males in both seasons and females in the dry season have a similar relationship to trees: positive preference at a small scale (intermediate preference for females), negative preference at a medium scale and positive preference again at a large scale (Fig. 7). Since the tree variable in Maputo is the proportion of closed woodland, and not divided into types, elephants might select certain tree types against others. For instance, elephants may prefer one type of tree at a small scale but select for clearings within another type, resulting in the opposing preference patterns at medium and large scales (see Fig. 2).

We also discovered that over-smoothing is a potential problem in performing these analyses. We initially extended the smoothing radius to >100 pixels (50 km), and found that there were often peaks (or even steadily improving) AIC values in this region. When smoothing occurs at this scale, environmental data tend to change monotonically across the entire landscape. In that case, there is a risk that any slight bias in the mean movement (i.e., any drift in the overall locations of animals), whatever the cause, will likely show up as a preference for, or avoidance of, that variable. This reflects the general truth that the spatial autocorrelation inherent in smoothed data will increase the degree of apparent correlations between variables, meaning that one must be very careful about assigning cause and effect. As our suburb/city example illustrates, when considering large-scale preferences, variables may act as proxies for something else. Because of these potential problems, we suggest using a maximum smoothing radius no more than twice the size of the choice radius of local movements. Smoothing only up to the choice radius precludes individuals from perceiving the environment at distances greater than they can travel.

We expected to see differing responses between male and female elephants, since females are typically part of a mixed herd containing juveniles that are less mobile than adults, while males are often solitary and can move larger distances (Smit, Grant & Whyte, 2007). However, while our data on males covers a greater spatial extent, differences in preference are quite small—even when males occupy regions for which we have no female data, their preferences remain similar. This suggests that our findings are reasonably robust, even though the elephant movements in this analysis do not encompass the entirety of each reserve’s landscape.

This study demonstrates that incorporating multiple spatial scales improves predictions of species habitat preferences, and as a consequence may dramatically alter landscape-wide maps of habitat quality. Discovering these habitat preferences helps identify the resources required by the species, at the correct scale, allowing wildlife managers to provide or restore them. The habitat preference maps help conservation planners ensure that enough habitats remain available, and accessible, for the target populations. For elephants, this is especially critical, given proposals that would allow protected populations greater freedom to roam (van Aarde, Jackson & Ferreira, 2006; van Aarde & Jackson, 2007).

Supplemental Information

Supplemental Information 1 Mathematica code used to smooth environmental landscape data and fit the conditional logistic model

Click here for additional data file.

Table S2 Results of the base-scale model selection process for each dataset

Click here for additional data file.

The findings and conclusions in this article are those of the author(s) and do not necessarily represent the views of the US Fish and Wildlife Service, the University of Pretoria, the Namibian Ministry of Environment, or the Peace Parks Foundation.

Additional Information and Declarations

Competing Interests

Author Contributions

Animal Ethics

Data Deposition

Stuart L. Pimm is an Academic Editor for PeerJ.

Andrew F. Mashintonio and Gareth J. Russell conceived and designed the experiments, performed the experiments, analyzed the data, contributed reagents/materials/analysis tools, wrote the paper, prepared figures and/or tables, reviewed drafts of the paper.

Stuart L. Pimm wrote the paper, reviewed drafts of the paper.

Grant M. Harris contributed reagents/materials/analysis tools, wrote the paper, reviewed drafts of the paper.

Rudi J. van Aarde contributed reagents/materials/analysis tools, reviewed drafts of the paper.

The following information was supplied relating to ethical approvals (i.e., approving body and any reference numbers):

The fieldwork was facilitated through permission from the Namibian Ministry of the Environment (Research permit number 580).

The following information was supplied regarding the deposition of related data:

Dryad: DOI 10.5061/dryad.nb5vc.

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
