# Peer review of "Data-driven discovery of the spatial scales of habitat choice by elephants"

_PeerJ, doi:10.7717/peerj.504_

## Round 0.1 · original submission · Minor Revisions

· Academic Editor

Minor Revisions

This research represents a very interesting approach to dissect how animals, and in particular elephants, integrate geo-spatial and ecological data to modulate its habitat selection. As whole, the paper is very well written using excellent English and complies with quality standards to be published in PeerJ. However, there are still some minor issues raised by the second reviewer that should be addressed by the authors before publication. The use of mixed models based on integrative kernels is a novel approach but it should better explained, including more mathematical formalism, as raised by the reviewers. Thanks a lot for considering PeerJ to publish your best research.

·

Basic reporting

Findings are very clearly reported and interpreted, the figures seem appropriate and the material covered is coherent

Experimental design

The originality of the findings lies in how multiple scales of habitat selection are assessed, and methods are clearly described

Validity of the findings

the assessment is statistically rigorous, and the findings are thoughtfully interpreted

Additional comments

A potentially influential factor not considered by the authors is time of day. I suspect that elephants might be less influenced by proximity to humans during the night than in the day, and that females might be affected more than males. Habitat influences could also vary depending on when the elephants sought water for drinking.

Reviewer 2 ·

Basic reporting

To the best of my knowledge, this submission meet appropriate scientific standards.
No evident inconsistencies or failures were detected in the process of reviewing this manuscript.

Experimental design

- Minor comments to improve the Introduction:
From line 34 to 54 a nice example is built using patterns of urban elements of a city to ilustrate the scope of the model. Certainly the analogy is useful to clarify ideas, but I believe it would be better to reformat that example to suit the real ecological system that will be analyzed later on.
That would be useful to gain better insight of the model as well as of the ecology of elephants at the same time, specially because the latter is largely discussed in the Results and Discussion sections.
Since this part of the manuscript is linked with a plot, and in order to avoid remaking the maps therein included, I would suggest to put this example in more general and abstract terms.

- Minor comments to improve Methods:
1) It would be valuable to have a longer explanation regarding the importance of the functional form of the smoothing kernels. At the heart of the research problem posed by the authors is the idea that animals integrate different environmental drivers at different scales. However, emphasizing that fact does not resolve the deeper problem on how that integration occurs. When smoothing data by averaging across pixels, we are assuming that the chosen kernel correctly approaches the manner in which the animals would do this integration of scale in the field. Obviously, it is not possible to test the validity of that statistical choice, but I think is important make an explicit justification regarding the chosen kerner, in order to make a meaningful interpretation of the model results.

2) The claims made from line 153 to 158 are too strong and may confuse readers with a background in mixed model, regarding the role of random factors. Therefore they should be downgraded or even removed from the sentence. The fact that all individuals showed the same kind of response to the environmental variables is not a valid reason to dismiss the inclusion of random factors. In fact, a model of random intercepts assume that groups show the same type of relationship between the predictor variables and the response, but displaced in the baseline.
This methodological comment is not necessary to accomplish the overall aim of the paper, so I recommend its removal.
In addition, parsimony for the random component can be tested by AIC, AICc and/or BIC, as long as a good null model is proposed.

3) I find counterintuitive the methodological step described between lines 187 to 200. When thinking about a modelling approach aimed to discovering the appropriate "scale" at which animals respond, discretizing that variable into three arbitrary scales seems at odds with the purpose of the model. I realize that at some point a convenient choice of scale must be done to operationalize the method, but I think that more compelling arguments must be risen to convince the reader that this discretization will not obscure the inferential process.

Validity of the findings

All the findings presented by the authors are compelling, show scientific rigurosity and the procedures have been described to a degree of detail that could allow other researchers reproduce the analysis and find the same results.

Additional comments

This is a good work of research that aims to develop an empirical approach to cope with the problem of scale in the context of habitat selection. Since this kind of issues lie in the cross-road of many ecological themes still unresolved (e.g., species distribution modeling, quantitative conservation ecology, statistical ecology, etc), the manuscript will be a good contribution to the literature.
I recommend to the authors pay attention to the comments made in the Experimental Design section, in order to sand down a few rough edges. The interactive nature of PeerJ will guarantee that any other aspects that went unseen by this reviewer, be exposed in a transparent open discussion.

---

## Round 0.2 · accepted · Accept

· Academic Editor

Accept

Thanks a lot for addressing the issues raised by the reviewers. This is a very interesting approach to systematise animal behaviour in the context of explicit spaces and therefore should be of broad interest for PeerJ audience. Once again congratulations for an excellent work and for considering PeerJ for the publication of your best research.